# Investigation on the Deformation and Failure Characteristics of Concrete in Dynamic Splitting Tests

**DOI:** 10.3390/ma15051681

**Published:** 2022-02-23

**Authors:** Xuan Xu, Li-Yuan Chi, Jun Yang, Nao Lv

**Affiliations:** 1State Key Laboratory of Explosion Science and Technology, Beijing Institute of Technology, Beijing 100081, China; yangj@bit.edu.cn; 2School of Civil Engineering and Architecture, Anhui University of Science and Technology, Huainan 232001, China; lvnao1990@163.com

**Keywords:** concrete, dynamic split, incubation time, digital image correlation, fracture trace, fragment size

## Abstract

The dynamic response behavior of concrete is constantly concerned because of seismic, impact and explosion events in the service of constructions. As a classic device for testing the dynamic mechanical properties of materials, the splitting Hopkinson pressure bar was used to carry out dynamic splitting tests on concrete in this paper. The variation of the dynamic tensile strength against the stress rate was fitted by the incubation time criterion. The full-field strain distribution on the observed surface of the specimen at the crack initiation stage was obtained by the digital image correlation (DIC) method. Morphological characteristics of the fragmentized process of concrete specimens in splitting processes were obtained by combining the image processing techniques and the FracPaQ. The size distribution of fragments of concrete specimens was obtained by sieving. The results show that the strain concentration zone and crack initiation appear along the loading direction through the center of the specimen. The secondary cracks initiated occurred at the contact end of the specimen, which expanded along the strain concentration zone and then interacted with the main crack. At the early stage of crack extension, the main crack dominates the normalized length of fracture traces in the horizontal direction. The normalized length of the vertical fracture trace increases with the main cracks opening width and the expansion of the secondary crack. The relationship between the length and angle of fracture traces in the dynamic splitting process of concrete conforms to the Gaussian function. Finally, the fragment sizes decrease with the stress rates of impact loads.

## 1. Introduction

Concrete is a common construction material used in many applications, such as construction, underground engineering, military bunkers, etc. During the service period, concrete may be subjected to dynamic loads within a wide range of strain rates, including earthquake, impact and penetration, explosion and blasting, etc. [1] As a typical tensile and compressive anisotropic material, the failure of concrete is usually determined by the tensile properties [2]. Therefore, it has important engineering and scientific significance to study the tensile properties of concrete materials under impact loading.

Generally speaking, the dynamic properties of materials can be determined by the split Hopkinson pressure bar (SHPB), which can provide dynamic loads with strain rates in the range of 10^−1^–10^4^ s^−1^. Xia and Yao [3] introduced the developing history and working principle of the SHPB system in detail. According to the one-dimensional stress wave theory, the stress–strain relationship curves can be obtained by measuring the strain signals on the incident and transmitted bars with strain gauges. The International Society for Rock Mechanics (ISRM) and the American Society for Materials and Testing (ASTM) [4,5] suggest the Brazilian test as the recommended method for testing the static tensile strength of rock, and researchers in Japan initially take it as the test method for the tensile strength of concrete [6]. Zhao and Li [7] studied the dynamic tensile properties of granite through a Brazilian splitting test with SHPB, which is cited by many researchers. Lv et al. [8] used the SHPB device to conduct dynamic splitting experiments on concrete specimens at different temperatures and analyzed the influence of temperature on the splitting strength and failure pattern of concrete. Guo et al. [9] conducted dynamic splitting experiments on high-strength concretes. It is found that the tensile strength increases with the strain rate, which shows an obvious strain rate enhancement effect. Hu et al. [10] studied the bonding behavior of concrete interface through a dynamic splitting experiment. The experimental results show that the strain rate has a significant influence on the failure pattern, compressive load–deformation curve, absorption energy and splitting tensile strength. The above research shows the feasibility of the splitting testing by the SHPB to determine the dynamic tensile strength of concrete materials.

In the past, many researchers proposed several expressions to describe the relationship between concrete tensile strength (*DIF*) and strain rate [11,12,13,14]. The formula proposed by CEB [15] concrete model specification divides the curve of the relationship between the *DIF* and the strain rate into two parts, namely the equation below the strain rate threshold and the subsequent equation. After the inflection point, different formulas can be found in the literature to describe the variation of *DIF* with the strain rate. In the experimental study of Ross et al. [11] and Tedesco et al. [16], the *DIF* of tensile strength of concrete changes linearly with the logarithm of strain rate, and there is a strain rate sensitive threshold at 1 s^−1^. However, Hao et al. recommend the use of bilinear [17] and high-order polynomial [18] to describe the increase in tensile strength of concrete with loading rates. Although these expressions are simple and intuitive, they are all phenomenological laws obtained by fitting experimental data. Kalthoff, Homma and Shockey proposed a special physical constant with time dimension, called incubation time [19,20,21], to describe the dynamic strength characteristics of brittle media. On this basis, Petrov [22] proposed a structural-temporal fracture criterion based on this theory, and this criterion can effectively predict the dynamic enhancement effect of rock and concrete-like materials [23,24].

The data measured in the SHPB system are strain signals of elastic bars by strain gauges. Strain gauges, as a standard mechanical measurement technique, can only measure an average deformation in a limited region. With the development of high-speed camera and optical measurement technology, non-contact full-field measurement methods were applied to capture strain distributions in the dynamic experiments of brittle materials such as rock and concrete [25]. Xu et al. [26] observed the effects of different loading boundaries on strain concentration evolution and tensile strength of concrete specimens by high-speed cameras combined with the DIC method. Okeil [27] revealed the difference in crack propagation between the reinforced surface inside the concrete and the surface of cement slurry through DIC analysis results. Combining DIC technology, Khan [28] studied the dynamic tensile properties of fiber-reinforced geopolymer composites (FRGC), the failure process and the crack opening displacement (COD) of different geopolymer specimens under different loading rates were obtained. According to previous studies, the DIC technique was found to be a widely used method for calculating the deformation of concrete materials.

When describing the dynamic mechanical behavior of concrete materials, the strength is not sufficient as a single criterion for evaluation. The initiation, propagation, penetration, and formation characteristics of cracks are equally important. In the dynamic splitting test, the DIC was satisfactory only at the early stage of crack initiation, although it could analyze the strain field distribution of the concrete surface based on images. The forming process of main cracks and secondary cracks in the specimen can be clearly captured by a high-speed camera. In order to quantitatively evaluate the morphology characteristics of the crack expansion, FracPaQ (V2.4, 2018, Aberdeen, UK) [29], a MATLAB-based open-source code, was used to identify and extract cracks on the surface of the concrete in the splitting process. In FracPaQ, the Hough transform method [30,31] is used to find collinear patterns of pixels in images and generate crack traces, then the distribution characteristics of crack, including direction, density, length, intensity and connectivity, can be statistically analyzed.

In this paper, the mechanical behaviors of concrete in the dynamic splitting test are investigated. The stress wave signals collected by the dynamic strain gauge are analyzed, and the variation of the concrete dynamic tensile strength against the stress rate is observed. The strain field of concrete specimens at the initiation of cracking was obtained by high-speed image analysis with the DIC technique. Fracture traces generated by FracPaQ are used to quantify the morphological characteristics of concrete specimens in the splitting process. Finally, the split fragments of concrete are sieved, and the statistics and the influence of the stress rate on fragment size distribution are revealed.

## 2. Experiments and Methods

### 2.1. Specimens Preparation and Static Test

The raw materials used for concrete specimen preparation are as follows: the cement was ordinary Portland cement (P.O42.5, Anhui Conch Group Co. LTD, Anhui, China), and relevant parameters are shown in Table 1. River sand (Anhui Conch Group Co. LTD, Anhui, China) with a density of 2600 kg·m^−3^, a maximum particle size of 5 mm and a fineness modulus of 2.60 was used as fine aggregate. The mineralogical composition of the river sand includes (weight percent): quartz of 59.8%; amphibole of 24.3%; albite of 9.7% and other minerals of 6.2%. The tap water, the chemical composition of which includes: chloride of 70.8 mg/L; sulfate of 31.2 mg/L; nitrate of 8.81 mg/L; and some other components of <1 mg/L, was applied. Crushed stone with a particle size of 5~20 mm was used as coarse aggregate. The polycarboxylic acid superplasticizer (Shengshi building materials, Guangzhou, China) was used as a water-reducer. The proportion of concrete mix is shown in Table 2.

When the thickness of the disk specimen was about half of its diameter, the influence of the inertia effect on test results could be eliminated. Therefore, a cylinder with a diameter of Φ 100 mm and a thickness of 50 mm was selected for the dynamic splitting specimen. The static Brazilian test was performed on cylindrical specimens with the same size of Φ 100 × 50 mm. This method was widely adopted by researchers to characterize the dynamic properties of materials [32,33]. Since the maximum particle size of aggregate is 20 mm and less than 31.5 mm, the cube specimen of 100 mm × 100 mm × 100 mm can be used for a static compressive test [34]. Concrete specimens are poured with the designed molds and cured in accordance with ASTM C192 [35] for 28 days. After curing, the specimen is ground on the two ends with a grinder, which requires that its non-parallelism is within 0.02 mm. The final length-diameter error of the specimen is within ±0.02 mm.

With a TBM machine (Sunstest, Guangzhou, China), static compressive and splitting tests were carried out as shown in Figure 1. During each splitting test, the specimen was mounted vertically between the upper and lower loading plates of the testing machine. In these static tests, the loading rate was controlled as 0.1 kN·s^−1^. In each group, three specimens were tested, and the average static compressive strength and tensile strength of concrete were 52.5 MPa and 3.0 MPa, respectively.

### 2.2. Principle of SHPB

The SHPB system mainly consists of a bullet, an incident bar, a transmission bar, a damper and a set of data acquisition systems, as shown in Figure 2. All the bars were made of steel with a Young’s modulus of 210 GPa and elastic wave velocity of 5190 m/s. The diameter of the bullet, incident bar and transmission bar are 100 mm, and the lengths are 600 mm, 4500 mm and 2500 mm, respectively.

In the SHPB system (Luoyang Liwei Technology CO., Ltd., Henan, China), the bullet is driven by an air gun using compressed nitrogen. Then the bullet strikes the incident bar, producing a stress wave (moving to the right) in the incident bar. When the stress wave reaches the interface between the concrete specimen and the incident bar, a part of the wave is reflected back to the incident bar to generate the reflected tensile wave. The remaining wave passes through the concrete specimen and transmits into the transmission bar. These waves can be recorded by the strain gauges on the incident bar and transmission bar.

Gomez’s study [36] shows that when the specimen is a homogeneous material, the equation used in static loading can also be used to calculate the dynamic tensile stress in the center of the specimen under dynamic loading conditions. However, concrete is a heterogeneous composite material. Through elastoplastic analysis and experimental measurement, Guo et al. [9] pointed out that the static elastic analysis can also be used to estimate the stress distribution in heterogeneous concrete specimens during dynamic splitting when the stress equilibrium condition is satisfied. Therefore, according to the one-dimensional stress wave theory, the indirect tensile stress in concrete specimens can be expressed as:(1)$$\sigma_{t}\left(t\right)=\frac{EA\left[\varepsilon_{i}\left(t\right)+\varepsilon_{r}\left(t\right)+\varepsilon_{t}\left(t\right)\right]}{\pi DB}$$
where *D* and *B* are the diameter and thickness of the specimen, respectively, and *E* and *A* are the elastic modulus and cross-sectional area of the compression bar, *ε_i_*(*t*), *ε_r_*(*t*) and *ε_t_*(*t*) are the incident wave, reflected wave and transmitted wave, respectively.

### 2.3. Generation of Speckle field for DIC Analysis

The DIC measurement technology (Mentel Co., Ltd., Brunswick, Germany) can calculate the displacement and strain information through the change of pattern or texture on the specimen surface. In the experiment, a speckle pattern with random gray distribution is required to be generated on the specimen surface, which is called a speckle field. Without affecting the mechanical behavior of the specimen, the speckle field varies with the deformation of the specimen [37]. A speckle field is usually prepared by artificial spray painting. The quality of spray speckle varies with the spraying process and surrounding environment. The digital speckle field generated by the software is used in this experiment, which can effectively solve the problems of large differences in speckle making and unstable quality.

A digital speckle field is generated by controlling the number of speckles, the coordinates of the center of the circle and the radius of the circle. The digital speckle field is generated by Equations (2)–(5) [37]:(2)$$\begin{array}{l} X_{i}=2\times i\times x_{i},\;i\in [1,n]\\ Y_{i}=2\times i\times y_{i},\;i\in [1,n] \end{array}$$
(3)$$\begin{array}{l} {X^{\prime }}_{i}=X_{1}+0.5\times f(r)\times a\\ {Y^{\prime }}_{i}=Y_{1}+0.5\times f(r)\times a \end{array}$$
(4)$$a=\frac{d}{2}\sqrt{\frac{\pi }{\rho }}$$
(5)$$n=\rho A/(0.25\cdot \pi d^{2})$$
where (*X*_1_, *Y*_1_) is the center coordinate of the first speckle, (*X_i_*, *Y_i_*) and (*X**′_i_*, *Y**′_i_*) are the center coordinate of the speckle in the speckle field with regular distribution and random distribution, respectively. *a* is the center distance of two speckles in the speckle field with regular distribution, and *ρ* is the duty ratio. The number of speckles n in the digital speckle field is related to the resolution A of the camera, which is determined by Formula (5). *f*(*r*) represents a pseudo-random function with an interval of (−*r*, *r*), and *r* is a random factor with a range of (0, 1).

In general, when the resolution of the camera is certain and the speckle diameter, random factor and duty ratio are determined, a digital speckle field in the form of a random distribution vector graph can be generated according to Equation (3). After several comparisons, the parameters of digital speckle are *r* = 0.15, *ρ* = 0.4, *d* = 1.3 mm. The generated digital speckle field and the specimen containing speckle are shown in Figure 3.

## 3. Results and Analysis

### 3.1. Typical Voltage Data and Stress Equilibrium Validation

The responses of concrete specimens S1 and S6 measured by the SHPB system under two different impact loads are shown in Figure 4 and Figure 5, respectively. Wherein, (a) is the typical voltage signal curve recorded by the strain gauge on the incident bar and transmission bar, which is divided into an incident wave, reflected wave and transmitted wave. It will be noted that the transmitted wave signal is quite weak compared to the incident or reflected wave. This is mainly because the splitting tensile strength of concrete is lower than its compressive strength. Therefore, a large part of the elastic stress wave is reflected along the incident bar as a tensile wave after the specimen is destroyed.

The dynamic tensile strength of concrete is affected by the stress rate, which can be obtained from the stress time history curve. As shown in Figure 4b, the stress changes approximately linearly between 70 μs and 125 μs. By linearly approximating this stretch stress–time curve, a constant loading rate of 103.1 GPa·s^−1^ can be obtained. The dynamic tensile strength of concrete is 9.64 MPa. Similarly, as shown in Figure 5b, when the loading rate is 423.1 GPa·s^−1^, the dynamic tensile strength of concrete is 23.64 MPa. In the process of dynamic loading, whether the specimen can achieve stress equilibrium is an important indicator for the validity of data processing. Figure 4c and Figure 5c are stress equilibrium checks in the case of eliminating the time lag between the three waves. According to the three-wave equilibrium theory, when the sum of incident stress and reflection stress equals transmission stress, the specimen is considered to be in stress equilibrium, i.e., In. + Re. = Tr. It can be seen that the sum of incident stress and reflection stress is close to transmission stress, indicating that stress equilibrium is approximately achieved in the SHPB test. To further verify the stress equilibrium, a stress equilibrium factor *η* is introduced. Its variation with time is used to reflect the loading state of concrete in the process of resisting impact load [38]. *σ*_A_ represents the sum of stress at the incident end, i.e., *σ_i_*(*t*) + *σ_r_*(*t*), *σ*_B_ represents the stress at transmission end *σ**_t_*(*t*), then the expression of *η* is written as:(6)$$\eta =\frac{2(\sigma_{A}-\sigma_{B})}{\sigma_{A}+\sigma_{B}}$$

As shown in Figure 4d, at the loading rate of 103.1 GPa·s^−1^, the stress wave reaches the stress equilibrium at 59 μs after repeated reflection in the specimen, and the stress equilibrium factor is approximately zero. When *t* = 225.6 μs, the stress equilibrium factor deviates from zero due to the curve fluctuation of *σ*_A_ and *σ*_B_ caused by the fracture of the concrete specimen. A similar phenomenon exists in the specimen with a loading rate of 423.1 GPa·s^−1^, but the difference is that the specimen loses stress equilibrium at 198.4 μs. Although the specimens lose their stress equilibriums at different times under the two loading rates, it could be seen that both specimens were in the unloading stage, which met the prerequisite conditions (i.e., stress equilibrium) of the SHPB experiment. Therefore, the experimental results are reliable. The contrast between Figure 4 and Figure 5 indicates a significant dynamic strength increase, although the specimens lose their stress equilibrium earlier at the higher stress rate loading.

### 3.2. Influence of Stress Rate on the Dynamic Tensile Strength of Concrete

The dynamic increase factor (DIF) is usually used to describe the increase in tensile strength of concrete materials at a higher loading rate, that is, the ratio of the dynamic strength to quasi-static strength, as shown in Equation (7):(7)$$DIF=\frac{f_{\mathrm{td}}}{f_{\mathrm{ts}}}$$

According to the structural-temporal criterion proposed by Petrov [22], the failure occurs if the stress state of a point *r* in the material meets the following conditions:(8)$$\frac{1}{\tau }\int_{t-\tau }^{t}\frac{1}{\delta }\int_{r-\delta }^{r}\sigma \left(r^{\prime },t^{\prime }\right)dr^{\prime }dt^{\prime }\leq \sigma_{0}$$
where, *τ* is the incubation time, which is the characteristic constant response to the load on the time scale and is not affected by specimen geometry, load pulse shape or loading mode. *δ* is a material constant, which is used to reflect the sensitive size of the material fracture zone, so it can characterize the characteristics of the fracture at a spatial scale. The function *σ*(*r*′, *t*′) (integrand function) represents the gradient of stress eigenvalues in space and time. Different physical quantities can be selected according to different criteria, such as compressive strength and tensile strength. *σ*_0_ corresponds to the static critical strength of the function. Assuming that the stress field near the fracture point only depends on time, Equation (8) of the incubation time criterion can be simplified and rewritten as:(9)$$\frac{1}{\tau }\int_{t-\tau }^{t}\sigma \left(t^{\prime }\right)dt^{\prime }\leq \sigma_{0}$$

If Equation (9) is used to study the relationship between the critical strength of materials and the stress rate, it should be assumed that the stress growth of materials can be considered linear at a certain stress rate until the peak of failure stress is reached [39,40]. So, it can be concluded that:(10)$$\sigma \left(t\right)=\sigma H\left(t\right)$$

In the formula, σ is the stress rate, and the stress in the specimen changes with time, as shown in Figure 4b and Figure 5b; *H*(t) is the step function. Substitute the above equation into Equation (9), and the instantaneous peak stress can be obtained:(11)$$\sigma^{*}=\sigma \left(t^{*}\right)=\left\{\begin{array}{c} \sigma_{0}+\dot{\sigma }\tau /2\;\left(t^{*}\geq \tau \right)\\ \sqrt{2\dot{\sigma }\sigma_{0}\tau }\;\left(t^{*}<\tau \right) \end{array}\right.$$

Obviously, for a low-stress rate, the instantaneous peak stress conforms to the condition *t** ≥ *τ* in Equation (11). For example, in the static process σ≈0, the peak failure strength is equal to the static strength.

The scattered points in Figure 6 are the dynamic tensile strength determined by the dynamic splitting test. The least square method was used to fit Equation (11) with the experimental data, and the incubation time of concrete was 96.2 μs. The prediction of dynamic tensile strength of concrete at different stress rates is shown as a continuous line in Figure 6. Their variation reflects that dynamic tensile strength enhancement is determined by quasi-static strength and incubation time, which allows gradient connection of materials from static to dynamic.

### 3.3. Deformation Characteristics of Concrete Measured by DIC

The digital image correlation (DIC) technique is a non-contact method to measure deformation. The full-field strain of DIC analysis is based on pixel subset matching of the camera captured image. In order to capture the strain field before the macroscopic crack appeared, a high-speed camera was arranged along the direction perpendicular to the circular section of the specimen. Images can be captured at a frame rate of up to 80,000 fps (frames per second) with a resolution of 256 × 256 pixels. To avoid damage to the equipment caused by the projectile of concrete fragments, the specimen is protected in a recycling box containing a polymethyl methacrylate (PMMA) shield. The first frame was captured by a high-speed camera as a reference image for DIC analysis. The frame with no obvious stress fluctuation in the calculated results was defined as 0 μs. Generally, cracks that expand along the direction parallel to the loading diameter are called primary cracks, and cracks that originate and expand in other parts are called secondary cracks.

Figure 7, Figure 8 and Figure 9 show the strain field distributions observed on the surface of S1, S6 and S7 of concrete specimens, and the incident end is on the right. Some local high strains can be observed before visible cracks appear, which lays a foundation for revealing new crack trajectories and initiation positions in the subsequent dynamic loading process. In the Brazilian splitting experiment, the results are valid only if the failure occurs along the compression diameter, so the strain distribution along the diameter is most important. During the evolution of the strain field, the strain distribution along the diameter of the specimen shows different behavior. Figure 7 shows the evolution process of the vertical strain field of the specimen at a loading rate of 103.1 GPa·s^−1^, corresponding to the typical results of waveform curves in Figure 4. After the arrival of the incident wave, obvious strain concentration areas appear at the two contact points between the specimen and the pressure bar, as shown in Figure 7b. As the loading continues, the tensile strain in the center of the specimen diameter begins to concentrate at *t* = 175 μs and develops continuously along the diameter to the two contact points. In the process of strain concentration zone expansion, the crack starts and develops along the strain concentration zone, leading to splitting failure. It is worth noting that although strain concentration occurs at the two contact points, the main crack starts from the center, and the strain expands from the center to both ends, forming a strain concentration area. Figure 8 shows the evolution process of the vertical strain field of the specimen at a loading rate of 423.1 GPa·s^−1^, which is a typical result of the waveform curve in Figure 5. At the initial stage of dynamic loading, the high strain migration with the direction of stress wave propagation shows obvious characteristics. The incident wave entered from the contact interface between the pressure bar and the specimen, and a high strain concentration was formed at the contact end and the right part of the specimen at 150 μs. Then, they move forward along the propagation direction of stress waves and form a strain concentration area along the whole loading diameter at *t* = 200 μs. With the increase of time, the high-strain area becomes larger and expands, and there is a strain concentration area with secondary at the end, as shown in Figure 8d–f. In this process, cracks start and expand simultaneously with the strain concentration region.

It can be seen from Figure 7 and Figure 8 that the deformation expansion process of the specimens is basically similar, which goes through the generation and expansion of the strain concentration area and finally runs through both ends of the specimen. The difference is that the high strain of specimen S1 first generates from both ends and center of the specimen, and extends from the center to both ends, while the high strain of specimen S6 first initiates from the right end of the specimen, then expands along the loading diameter to the other end, and finally runs through the whole specimen. This is different from the standard Brazilian splitting failure mode, which may be because the stress concentration effect at the loading end is enhanced due to the high-stress rate, or there is damage at the right end of the specimen before loading or the weak surface of the structure, which is damaged first when subjected to dynamic load [26]. In addition, the difference between the test results under these two stress rates also lies in the penetration time of the strain concentration zone or the generation time of the visible main crack. Within the observation range, the penetration time of the strain concentration zone of the S1 specimen is 225 μs, and that of the S6 specimen is 200 μs. This is basically consistent with the time when the specimen loses stress equilibrium in Figure 4d and Figure 5d. This indicates the generation of cracks caused the specimen to lose its stress equilibrium.

As shown in Figure 9, the evolution process of the strain field of the S7 specimen is different from S1 and S6. Although the initiation location and propagation mode of the cracks in the latter two specimens were different, a single main crack finally penetrated through the specimen and split it, while multiple cracks appeared in the S7 specimen. At 150 μs, two strain concentration zones are generated inS7 specimen. Although the strain concentration mode is parallel to the horizontal loading direction, the two regions are not located in the same line. As the loading process continues, strain concentration areas of the two parts expand to both ends and soon reach the end which is close to each other. However, when it expands to the other end, the development is relatively slow, and the two strain concentration areas are not connected until 250 μs, as shown in Figure 9f. As can be seen from the expansion process of strain concentration area in Figure 9c–f, with the loading process proceeding. The strain concentration pattern deviates from the horizontal direction.

Theoretically, for the direct tensile experiment, the tensile stress in the cross-section is a constant uniform field, and the geometric center of the disk is the only point that meets the tensile failure condition, and its value is equal to the uniaxial tensile strength. However, the compressive stress at the loading end is much greater than the tensile stress under plate loading, which may lead to collapse or shear failure at the loading end. In addition, crack initiation of concrete materials is also easy to appear in the pores, cracks or multiphase interface stress concentration of concrete disk specimens, so the location of crack initiation is relatively random. It can be seen that dynamic Brazilian splitting does not necessarily start from the center, and the position of initiation is closely related to the structure of the specimen.

### 3.4. Fragmentized Process of Typical Specimens

In the above section, the strain field of the specimen was observed by DIC technology before the visible crack appeared. The local high strain propagation process, to a certain extent, predicted the development trajectory of the crack in the subsequent dynamic loading process. With the opening of the main crack on the surface of the specimen, the secondary crack mainly caused by shear failure appears gradually at the contact point between the specimen and the pressure bar. The process of primary crack opening and secondary crack initiation and propagation in typical concrete splitting results is shown in Figure 10, Figure 11 and Figure 12. The image captured by a high-speed camera is binarized to obtain the pixel information of the crack. Then, pixel information was used to measure the width of the main crack at different moments, as shown in Figure 10, Figure 11 and Figure 12.

Two types of cracks are generally observed from Figure 10, Figure 11 and Figure 12: tensile and shear. The tensile crack is almost parallel to the loading direction, while the shear crack is inclined to the loading direction. Combined with the evolution of the strain field in Figure 7, the cracks of the S1 specimen spread from the center to both ends. At 300 μs, the main cracks were clearly visible, and some minor secondary cracks were generated at both ends of the specimen. With the increase of time, the width of the primary crack increases, and the direction of the secondary crack gradually deviates from the primary crack in the process of expansion. Different from the crack propagation mode of the S1 specimen, it can be seen from the evolution of the strain field of the S6 specimen in Figure 8 that the crack of the S6 specimen expands from the right loading end to the left constrained end. With the increase of time, a compression fracture zone is formed at the loading end and the fracture zone is constantly expanding, while secondary cracks occur at the constrained end. Compared with the S1 specimen, a fracture zone may be generated at the loading end and the width of the main crack increases when the stress rate increases. This is because the friction force at the loading end is obvious with the increase of stress rate. The stress situation in this area becomes complex, and a large number of microcracks expand and connect at the same time, eventually leading to the generation of fracture zone, which also enhances the mechanical properties of the specimen under dynamic load. Figure 12 shows the crack growth process of the specimen at the stress rate of 428.3 GPa·s^−1^. It can be clearly seen that the width of the main crack far from the specimen center is smaller than that of the main crack at the specimen center, indicating that the main crack expands and runs through from the specimen center to the loading point, which is similar to the failure characteristics of S1 specimen and is a typical failure characteristic of Brazil test. Shear cracks all occur after the main crack expands to the loading point, which is at the end of crack propagation and reflects the stress mutation. The crack propagation process of S1, S6 and S8 specimens indicates that the initiation location of secondary cracks is basically at the end of the specimen. After the primary crack cracks, the high level of stress region in the specimen extends to the end of the specimen, resulting in the edge cracking of the specimen and the secondary cracks. It can be considered that the secondary crack expands along the edge of the strain concentration zone from the end and then crosses and merges with the primary crack. Therefore, cracks near the center of concrete are mainly caused by tensile failure of the axial primary crack parallel to the loading direction, while other cracks are caused by shear failure of secondary cracks resulting from further compression. According to the main crack widths at different times, the average opening velocities of the main crack were 6.37 m/s, 10.89 m/s and 10.95 m/s, respectively, at the stress rates of 103.1 GPa·s^−1^, 423.1 GPa·s^−1^ and 428.3 GPa·s^−1^ in the observation period.

### 3.5. Morphological Characteristics of Splitting Fragmentation

Based on the pixel information of binary images, FracPaQ can use the Hough transform method to find collinear patterns of pixels in the image and produce fracture traces. The fracture trace is a line segment whose endpoints are distributed along the crack contour. In other words, the cracks in the binary image are filled with fracture traces. The number and length of fracture traces can be influenced by adjusting the number of Hough peaks and the Hough threshold. Figure 13, Figure 14 and Figure 15 are analysis results of FracPaQ based on the crack binary images in Figure 10, Figure 11 and Figure 12, respectively. The number of Hough peaks is set as 3000, and the Hough threshold is set as 0.1. In these figures, the cartesian coordinate system is established, with the contact end of the transmission bar and the specimen as the origin, the direction of the bar is X-axis, and the direction perpendicular to the bar is Y-axis. Therefore, we can quantitatively characterize the crack by calculating the length and direction of the fracture traces. The normalized length is defined as the ratio of the length of a fracture trace to the sum of the length of all fracture traces at the current time. The distribution of the deviation angle of the normalized length of the fracture trace along the Y-axis was statistically analyzed, and a rose diagram reflecting the relationship between them was obtained, as shown in Figure 16. The Y-axis deflection angles are calculated from both clockwise (0° to 180°) and counterclockwise (180° to 360°), respectively, so these rose diagrams are center-symmetric. At the early stage of crack propagation (Figure 16 (a) 300 μs, (b) 300 μs, and (c) 175 μs), the normalized length of fracture traces corresponding to the main crack in the horizontal direction (Y-axis deflection angle is 90° and 270°, respectively) is dominant. With the increase of the opening width of the main crack and the expansion of the secondary crack, the fracture track line in the vertical direction gradually increases, which means that the values corresponding to the normalized lengths with Y-axis rotation angles of 0 and 180° gradually increase. Therefore, the ability of FracPaQ to characterize the crack change during specimen failure is verified.

In order to further describe the fracture characteristics of dynamic splitting of concrete specimens, the length distribution of fracture traces at Y-axis deviation angle (clockwise, 0~180°) of typical results under three stress rates were obtained, as shown in Figure 17. The distribution characteristics of the scattered points on the outer edge are close to Gaussian function, which can be expressed as:(12)$$y=y_{0}+Ae^{-\frac{{\left(x-x_{c}\right)}^{2}}{2w^{2}}}$$
where *y*_0_ is the offset of the X-axis, *x*_c_ is the center of symmetry, *w* is the width, and *A* is the amplitude.

Then, Equation (12) is used to fit the length scatter data of specimens S1, S6 and S8 at different times, and the fitting parameters are shown in Table 3. The results show that the relationship between the length and angle of fracture traces conforms to the Gaussian function, and the fitting results have a good correlation. As can be seen from the Gaussian function curve, the initial curves of the three specimens are large in width and small in amplitude. With the increase of time, the curves show a trend of decreasing in width and increasing in amplitude. In addition, *y*_0_ also increases with the increase of time, and more and more scattered points are distributed under the curve, which indicates that the crack opening width is larger, and more fracture track lines are filled with the increase of time. This corresponds to the pattern shown in the rose diagram in Figure 16. At the early stage of the crack propagation process, the normalized length of the fracture traces corresponding to the crack in the vertical direction (Y-axis deflection angle is 0° and 180°, respectively) account for a very small proportion. With the increase of time, the crack expands further, and the normalized length of the fracture track in the vertical direction increases gradually. In conclusion, the Gaussian function is used to fit the length and angle of fracture traces, which can characterize the fragmented process during the dynamic splitting of the concrete specimen.

### 3.6. Fragments Size Distributions of Dynamic Splitting Concrete Specimens

The typical failure pattern of the specimen in the dynamic splitting test is split into two halves along the loading direction. When the stress rate is large, there may be triangular failure zones at both ends of the specimen. However, visual estimates showed that the remaining fragments were too similar in size to discern a pattern. Therefore, a standard set of screens ranging in size from 0.045 mm to 20 mm were used to screen debris. Figure 18 shows the fractal characteristics of S2 and S5 specimens at stress rates of 150.3 GPa·s^−1^ and 378.5 GPa·s^−1^, respectively. The fragments with an aperture of more than 20 mm show a typical dynamic splitting failure pattern. The S2 specimen is symmetrically divided into two parts, and the fracture surface is relatively smooth compared with the S5 specimen. When the stress rate is large and the crack expands in concrete and meets the aggregate particle, the crack does not change the propagation direction in the mortar matrix but directly passes through the aggregate particle. This results in the fracture of aggregate particles and a relatively straight section. The stress rate of the S5 specimen is higher than that of S2. Under normal circumstances, the section of the S5 specimen should be smoother. However, it can be seen from Figure 18b that there is also a debonding phenomenon between aggregate and mortar (red circle) on the S5 specimen, except for some broken aggregate particles. This may be caused by the structure of the specimen itself. During the hardening process of cemented materials, a large number of micro-cracks and micro-holes will be formed in the specimen, which increases the randomness of the crack initiation location, propagation path and failure mode of the specimen. The fragment size distribution of specimen S5 from 0.045 mm to 0.18 mm is significantly larger than that of the S2 specimen, and this part of the fragment comes from the contact ends of the specimen. Due to the concentrated stress at both loading ends, the triangular crushing zone exists in traditional plate loading, and the crushing level and the crushing zone gradually increase with the stress rate.

In order to describe the fragments size distribution of concrete specimens under different stress rates, the Swebrec function with three parameters [41] is used to fit the sieving results. The function is denoted by *P*(*x*), which is the mass fraction of fragments passed through a sieve size of *x*:(13)$$P(x)=\frac{1}{1+{\left[\ln(x_{\max}/x)/\ln(x_{\max}/x_{50})\right]}^{b}},0<x\leq x_{\max}$$
where, *x*_max_ is the maximum size (*P*(*x*_max_) = 100%), *x*_50_ is the median size (*P*(*x*_50_) = 50%), and b is the fluctuation parameter. Volume-based particle size (*x* = $2\sqrt[{3}]{3V/4\pi }$, *V* is the fragment volume) is often used to characterize irregularly shaped fragments in sieving analysis. The value of *x*_max_ can be determined by directly measuring the size of the maximum fragment. Since the specimen is mostly divided into two parts in the dynamic splitting test, *x*_max_ can be set as 50 mm.

Due to the different distribution characteristics of concrete in the dynamic splitting test, the distribution rules are predicted based on the data points (11 data points) of the fine pieces (less than 20 mm). The screening data set and fitting curve are shown in Figure 19, and the fitting parameters are shown in Table 4. Through the Swebrec function, the fitting curves of fine particle size of concrete under different stress rates have a good correlation. As the mass ratio of fragments over 20 mm is more than 50%, it can be seen that the median size *x*_50_ under different stress rates is between 42 mm and 45 mm, with little difference. Figure 19 shows that with the increase of stress rate, the mass proportion of fragments passing increases. It indicates that the fragmentation decreases with the increase of stress rate.

## 4. Conclusions

In this paper, the SHPB system was used to carry out the dynamic splitting test on concrete under different loading rates. Based on the experimental results the incubation times of concrete specimens with 100 mm diameter and 50 mm high were determined, the deformation characteristics of the cracking initiation stage were revealed by the DIC technique, and the dynamic splitting failure process was described by the Gaussian function. The detailed conclusions are as follows:(1)Based on the SHPB test results, the rate-dependent effect of concrete was determined by using the incubation time criterion, and the incubation time was 96.2 μs. The dynamic tensile strength enhancement is determined by the quasi-static strength and incubation time.(2)The DIC technique was used to obtain the full-field strain of the observed surface at the main crack initiation stage. In addition, multiple main cracks may appear to run through the specimen, and the time of the main cracks running through the specimen is consistent with the time of the stress balance factor curve deviating from zero.(3)By using image processing technology, the crack information of concrete specimens in the fracturing is extracted. According to the binarization results, the average opening velocity of the main crack increases with the increase of stress rate. The initiation of secondary cracks occurred at the end of the specimen, which expanded along the edge of the strain concentration area and then crossed and merged with the main crack.(4)The crack pixel information is converted into fracture trace coordinates by FracPaQ. At the early stage of crack initiation, the normalized length of fracture traces corresponding to the main crack in the horizontal direction is dominant. The normalized length of the vertical fracture path increases with the increase of the main crack opening width and the expansion of the secondary crack. The relationship between the length and angle of fracture traces of dynamic splitting specimens at each moment in the process of crushing conforms to the Gaussian function.(5)The Swebrec function was used to fit the sieving results of dynamic Brazilian concrete specimens. The results showed that with the increasing stress rate, the percentage of mass passing increased, which meant that the fragment size decreased with the increase of stress rate.

## Figures and Tables

**Figure 1 materials-15-01681-f001:**
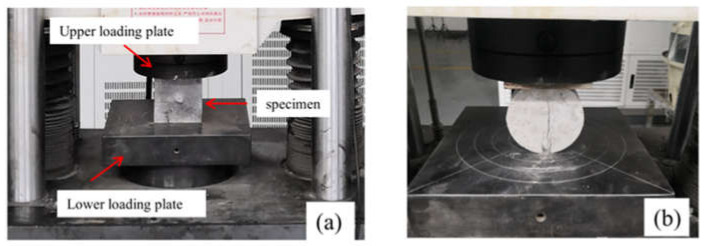
Static test: (**a**) compression; (**b**) split.

**Figure 2 materials-15-01681-f002:**
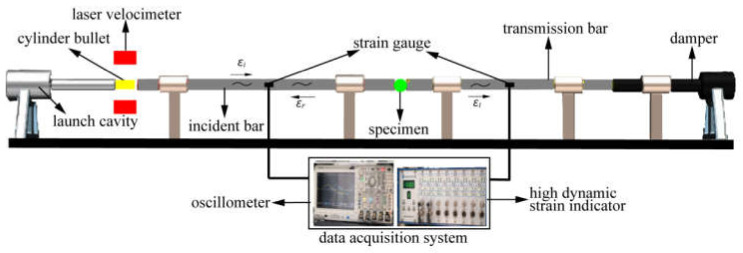
Schematic diagram of SHPB system.

**Figure 3 materials-15-01681-f003:**
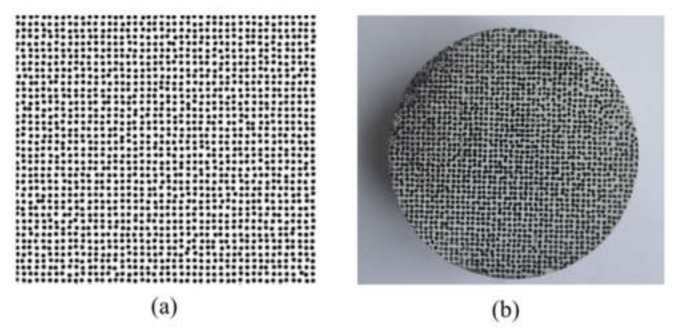
(**a**) Speckle field and (**b**) specimen.

**Figure 4 materials-15-01681-f004:**
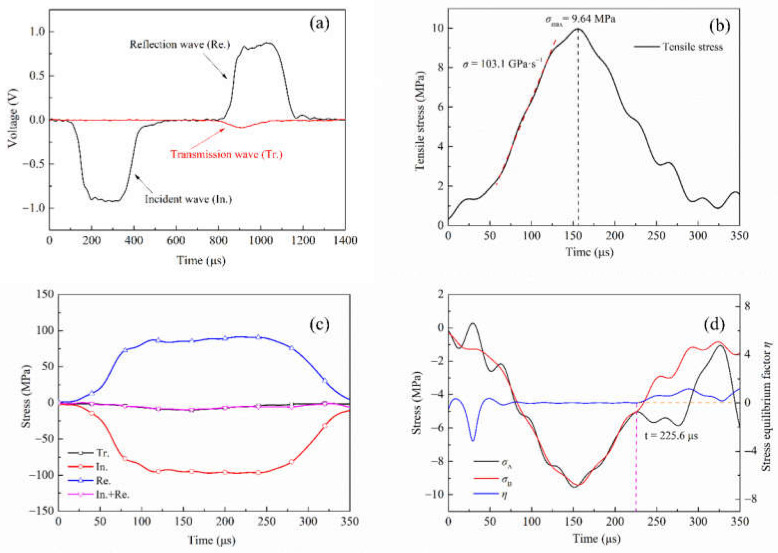
Experimental results of S1 specimen. (**a**) Raw signals of Brazilian test. (**b**) Determination of dynamic tensile strength and loading rate. (**c**) The incident (In.), reflected (Re.), transmitted (Tr.) and superposed (In. + Re.) waves. (**d**) Verify stress equilibrium.

**Figure 5 materials-15-01681-f005:**
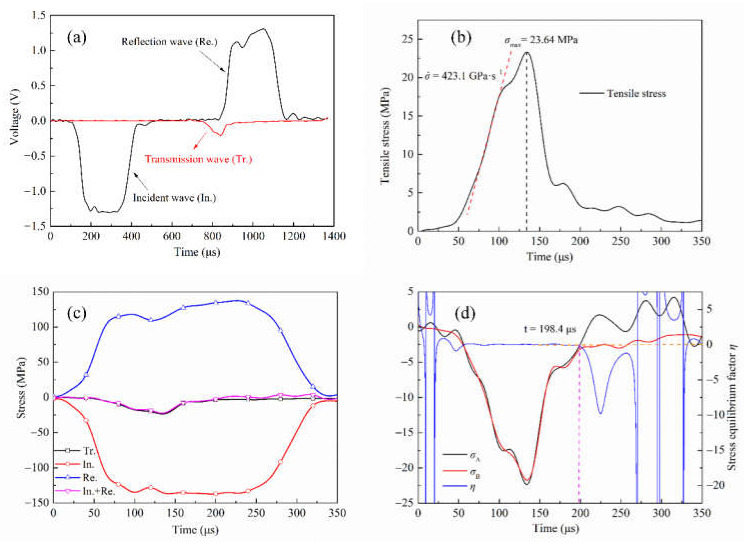
Experimental results of S6 specimen. (**a**) Raw signals of Brazilian test. (**b**) Determination of dynamic tensile strength and loading rate. (**c**) The incident (In.), reflected (Re.), transmitted (Tr.) and superposed (In. + Re.) waves. (**d**) Verify stress equilibrium.

**Figure 6 materials-15-01681-f006:**
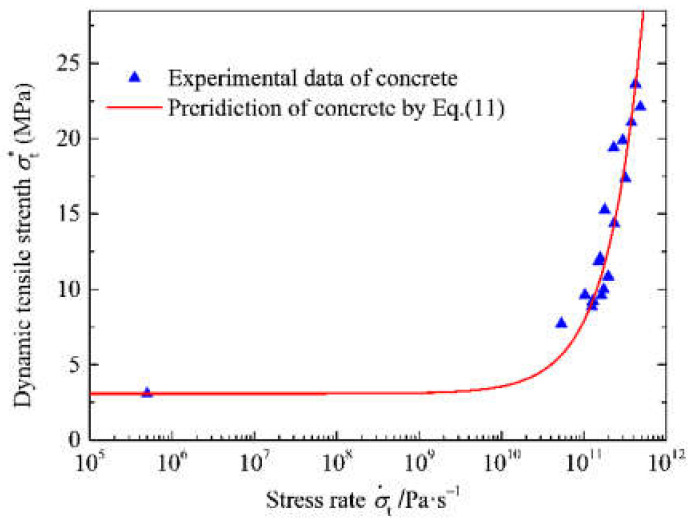
Relationship between dynamic tensile strength and stress rate.

**Figure 7 materials-15-01681-f007:**
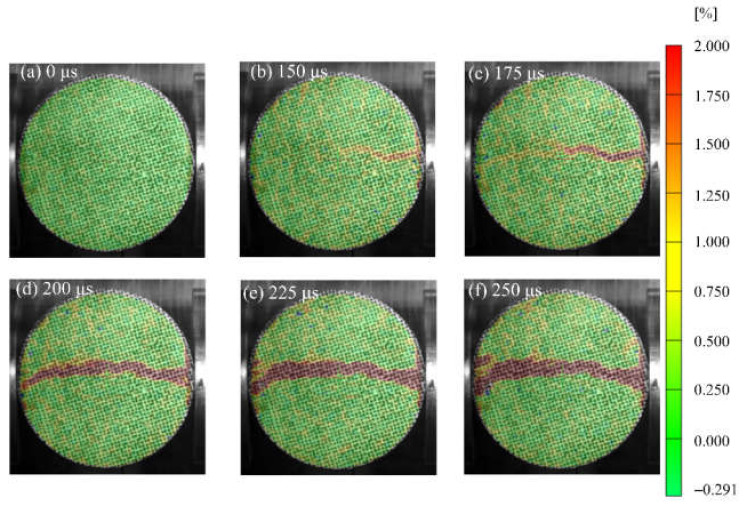
Full-field max-principal strain evolution of S1 specimen in dynamic splitting process: (**a**) 0 μs; (**b**) 150 μs; (**c**) 175 μs; (**d**) 200 μs; (**e**) 225 μs; (**f**) 250 μs.

**Figure 8 materials-15-01681-f008:**
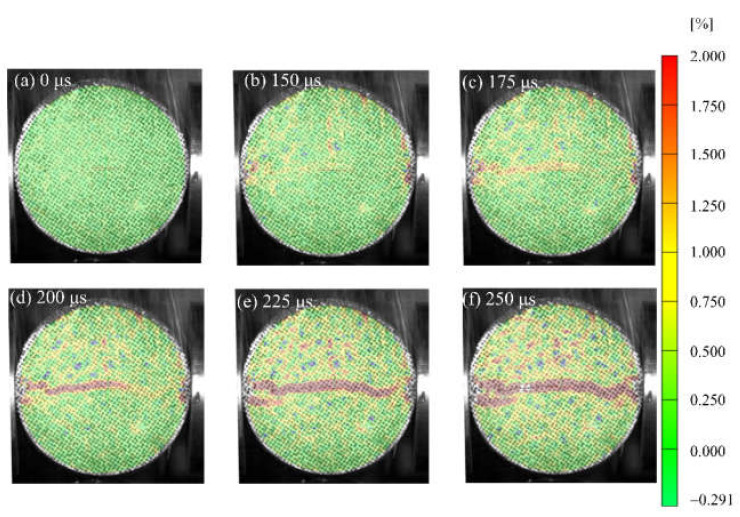
Full-field max-principal strain evolution of S6 specimen in dynamic splitting process: (**a**) 0 μs; (**b**) 150 μs; (**c**) 175 μs; (**d**) 200 μs; (**e**) 225 μs; (**f**) 250 μs.

**Figure 9 materials-15-01681-f009:**
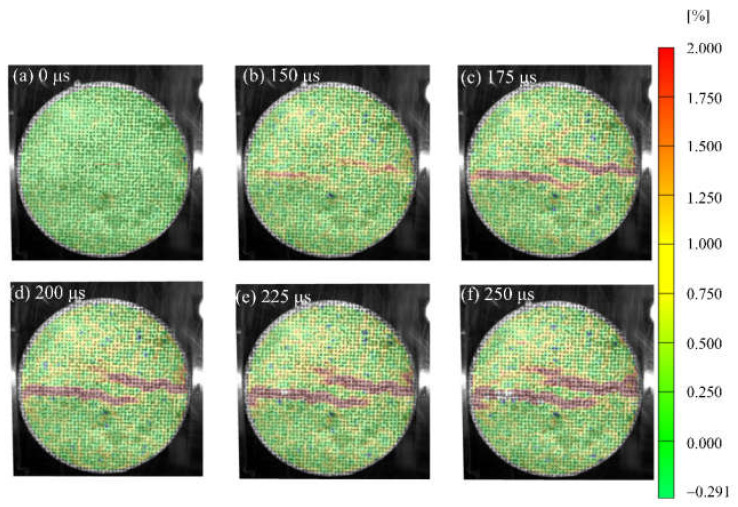
Full-field max-principal strain evolution of S7 specimen in dynamic splitting process: (**a**) 0 μs; (**b**) 150 μs; (**c**) 175 μs; (**d**) 200 μs; (**e**) 225 μs; (**f**) 250 μs.

**Figure 10 materials-15-01681-f010:**
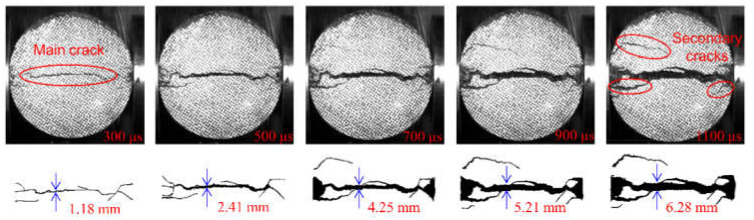
Fragmentized process of S1 specimen.

**Figure 11 materials-15-01681-f011:**
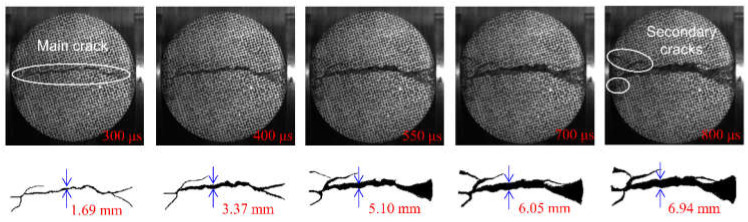
Fragmentized process of S6 specimen.

**Figure 12 materials-15-01681-f012:**
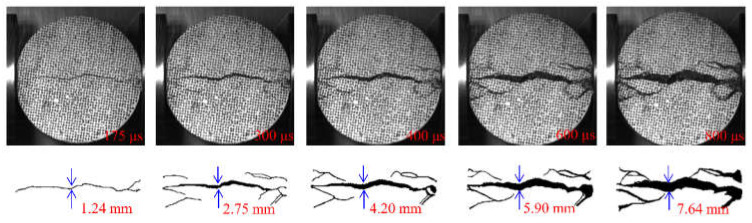
Fragmentized process of S8 specimen.

**Figure 13 materials-15-01681-f013:**
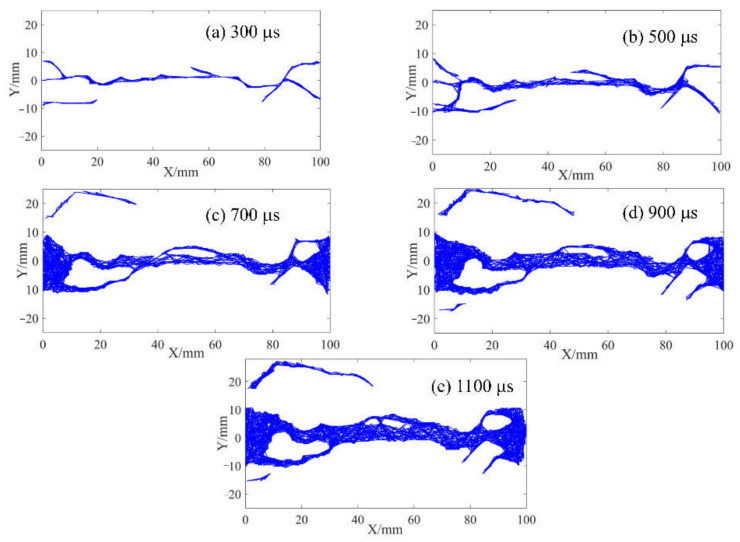
Filling results of fracture traces of S1 specimen: (**a**) 300 μs; (**b**) 500 μs; (**c**) 700 μs; (**d**) 900 μs; (**e**) 1100 μs.

**Figure 14 materials-15-01681-f014:**
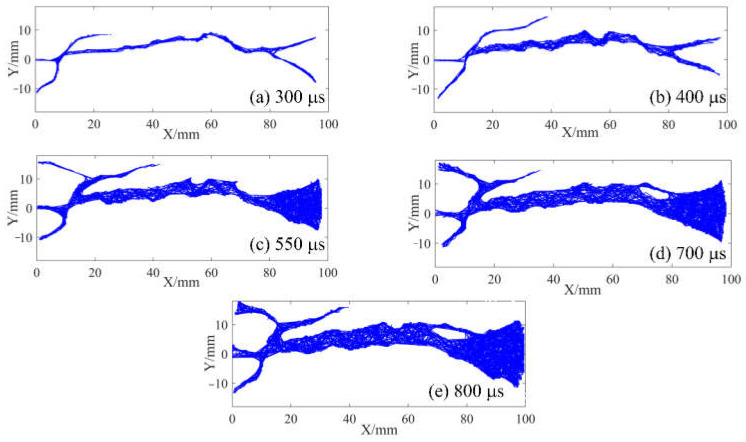
Filling results of fracture traces of S6 specimen: (**a**) 300 μs; (**b**) 400 μs; (**c**) 550 μs; (**d**) 700 μs; (**e**) 800 μs.

**Figure 15 materials-15-01681-f015:**
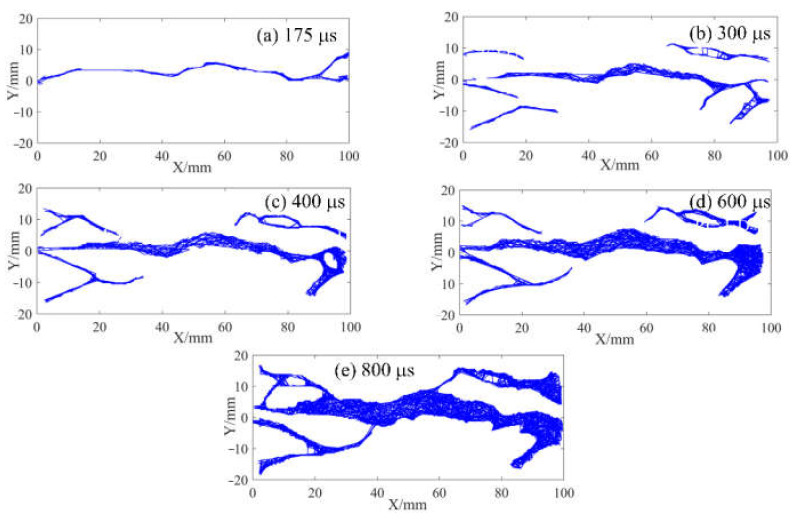
Filling results of fracture traces of S8 specimen: (**a**) 175 μs; (**b**) 300 μs; (**c**) 400 μs; (**d**) 600 μs; (**e**) 800 μs.

**Figure 16 materials-15-01681-f016:**
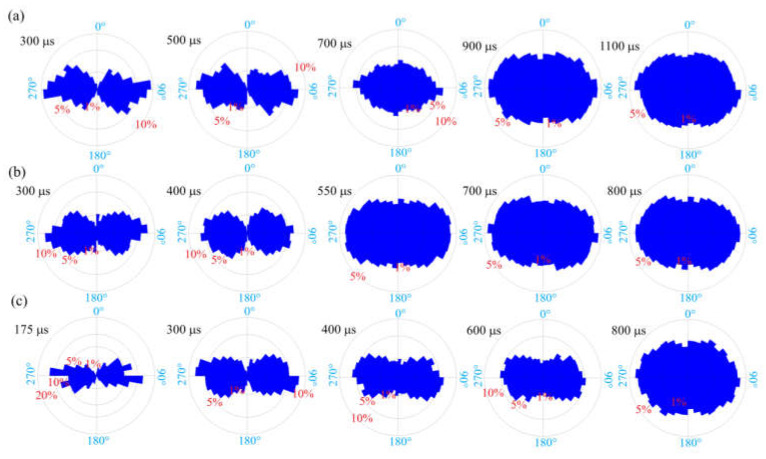
Rose diagram of normalized length of type specimens (**a**) S1, (**b**) S6 and (**c**) S8.

**Figure 17 materials-15-01681-f017:**
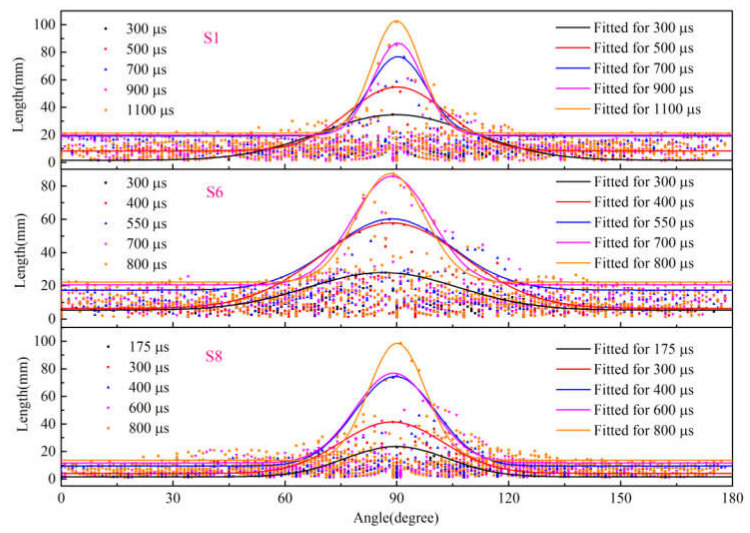
Scatters distribution and the Gaussian function fitted results of S1, S6 and S8 specimens at different times.

**Figure 18 materials-15-01681-f018:**
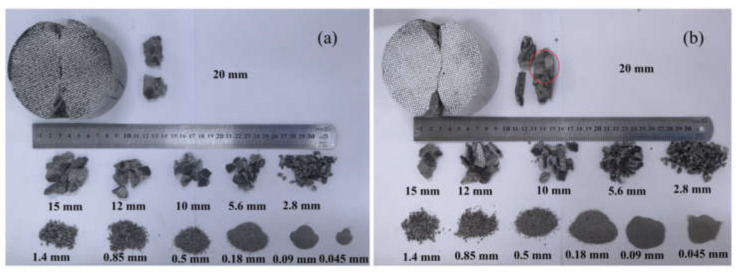
Sieving results of typical specimens: (**a**) S2; (**b**) S5.

**Figure 19 materials-15-01681-f019:**
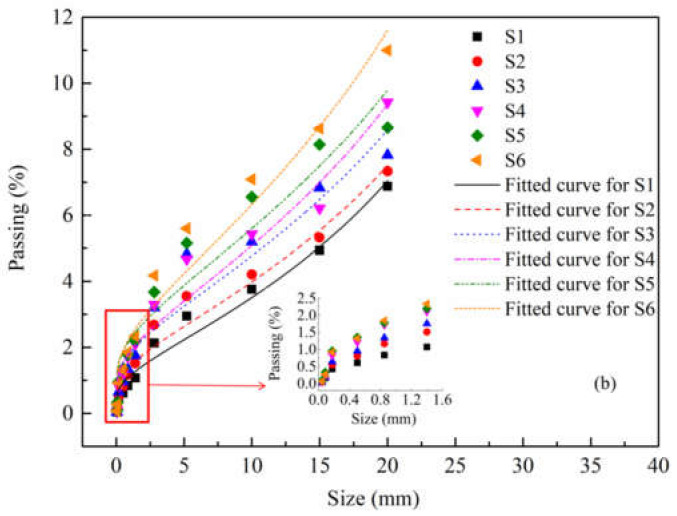
Sieving data and fitted curves of splitting fragments under different stress rates of S2 and S5 specimens.

**Table 1 materials-15-01681-t001:** Related parameters of cement.

Density (kg/m^3^)	BET (m^2^/g)	Chemical Composition (wt.%) (XRF)					
		CaO	SiO_2_	Al_2_O_3_	Fe_2_O_3_	SO_3_	MgO
1910	1.477	31.31	1.94	0.9	0.23	43.49	0.29

**Table 2 materials-15-01681-t002:** The mixing proportion of concrete.

Cement	Water	Fine Aggregate	Coarse Aggregate	Superplasticizer
495 kg/m^3^	167 kg/m^3^	544 kg/m^3^	1251 kg/m^3^	10.4 kg/m^3^

**Table 3 materials-15-01681-t003:** Fitted parameters of the Gaussian function.

Specimen	Time (μs)	*y* _0_	*x* _c_	*w*	*A*	*R* ^2^
S1	300	1.56	89.82	20.51	33.13	0.942
	500	8.36	90.11	13.15	46.47	0.987
	700	19.22	90.37	7.73	57.78	0.947
	900	19.73	90.06	6.75	66.23	0.970
	1100	21.37	89.94	6.54	81.57	0.917
S6	300	5.40	86.29	20.18	22.61	0.891
	400	6.43	88.58	19.27	51.48	0.987
	550	17.43	88.71	16.00	42.88	0.948
	700	20.51	88.67	10.64	65.35	0.878
	800	22.19	88.29	9.19	65.38	0.884
S8	175	1.56	89.88	13.79	22.12	0.926
	300	4.12	88.69	13.69	37.42	0.958
	400	9.46	89.61	10.95	65.14	0.952
	600	11.73	89.13	10.34	65.22	0.917
	800	13.69	90.60	7.58	84.96	0.868

**Table 4 materials-15-01681-t004:** Fitted parameters of the Swebrec function.

Specimen	*x*_max_ (mm)	*x*_50_ (mm)	*b*	*R* ^2^	Stress Rate (GPa·s^−1^)
S1	50	44.07	1.30	0.9673	103.1
S2	50	44.73	1.19	0.9512	150.3
S3	50	44.78	1.12	0.9115	198.6
S4	50	43.86	1.17	0.9441	318.8
S5	50	44.52	1.07	0.9145	378.5
S6	50	42.41	1.18	0.9459	423.1

## Data Availability

Not applicable.

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
