# Peer review of "Investigation on the Deformation and Failure Characteristics of Concrete in Dynamic Splitting Tests"

_materials, 2022, doi:10.3390/ma15051681_

Round 1
Reviewer 1 Report
Very nice scientific article. I evaluate positively the experimental test. Its results are beneficial. The article is well organized. Both the abstract and the introduction have the necessary requisites. I consider the references to be sufficient. The research methods are scientific and correct. I did not find any formal errors in the article. Perhaps just such a trifle that Ludolf's number pi is also written in vertical letters. It would be beneficial to be able to increase the contrast of Figures 4 and 5. I recommend publishing the article.
Author Response
Response to Reviewer 1 Comments
Dear Editors and Reviewers:
Thanks for your letter and for the reviewers' comments concerning our manuscript entitled "Investigation on the deformation and failure characteristics of concrete in dynamic splitting tests" (ID: materials-1594639). The comments are all valuable and helpful for revising and improving our paper, as well as the important guiding significance to our research. We have studied the comments carefully and have made corrections which we hope are met with approval. Revised portions are marked in red in the paper (Track Changes Version). The main corrections in the paper and the response to the reviewers' comments are as following:
Point 1: Very nice scientific article. I evaluate positively the experimental test. Its results are beneficial. The article is well organized. Both the abstract and the introduction have the necessary requisites. I consider the references to be sufficient. The research methods are scientific and correct. I did not find any formal errors in the article. Perhaps just such a trifle that Ludolf's number pi is also written in vertical letters. It would be beneficial to be able to increase the contrast of Figures 4 and 5. I recommend publishing the article.
Response 1: Thanks for your comment. The symbol of Pi in Eqs. (1), (4) and (5) are modified to be in vertical letters.
At the end of the preceding paragraph of Fig. 4, we added the sentence about the contrast between them: "The contrast between Figs. 4 and 5 indicates a significant dynamic strength increase, although the specimens lose their stress equilibrium earlier at the higher stress rate loading."

Reviewer 2 Report
Dear authors
your work is well designed and valuable, but the structure shoudl be improved in order to improve understandability. For example, there are parts that should fit better in the introduction, and the abstract needs a review.
Please take into account my suggestions in the attached document.
Regards.

Author Response
Dear Editors and Reviewers:
Thanks for your letter and for the reviewers' comments concerning our manuscript entitled "Investigation on the deformation and failure characteristics of concrete in dynamic splitting tests" (ID: materials-1594639). The comments are all valuable and helpful for revising and improving our paper, as well as the important guiding significance to our research. We have studied the comments carefully and have made corrections which we hope are met with approval. Revised portions are marked in red in the paper (Track Changes Version). The main corrections in the paper and the response to the reviewers' comments are as following:
Dear authors
your work is well designed and valuable, but the structure should be improved in order to improve understandability. For example, there are parts that should fit better in the introduction, and the abstract needs a review.
Please take into account my suggestions in the attached document.
Regards.
Point 1: Abstract
Please give a brief introduction on the State of the Art in the abstract, it is useful to introduce the reader to the topic.
Response 1: Thanks for your comment. We accept the reviewer's comment and revise the first sentence of the abstract as: "The dynamic response behaviour of concrete is constantly concerned because of seismic, impact and explosion events in the service of constructions. As a classic device for testing the dynamic mechanical properties of materials, the splitting Hopkinson pressure bar was used to carry out dynamic splitting tests on concrete in this paper."
Point 2: Page 2 Line 93
Typo "preparetion".
Response 2: Thank you for pointing out, we have corrected the typo.
Point 3: Page 2 Line 96
Mineralogical composition of the river sand?
Response 3: Thanks for your comment. We add the mineralogical composition of the river sand in the first paragraph of Section 2.1. "The mineralogical composition of the river sand includes (weight percent): quartz of 59.8%; amphibole of 24.3%; albite of 9.7% and other minerals of 6.2%."
Point 4: Page 2 Line 97
Chemical composition of the tap water?
Response 4: Thanks for your comment. We add the chemical composition of the tap water in the first paragraph of Section 2.1. "The tap water, whose chemical composition includes: chloride of 70.8 mg/L; sulfate of 31.2 mg/L; nitrate of 8.81 mg/L and some other components of <1 mg/L, was applied."
Point 5: Page 7 Line 227
Please cite some of these researchers.
Response 5: Thanks for your comment. We added the following references for this sentence.
- Malvar, L.J.; Ross, C.A. Review of strain rate effects for concrete in tension. ACI Mater. J. 1998, 95, 735–739.
- Brara, A.; Klepaczko, J.R. Experimental characterization of concrete in dynamic tension. Mech. Mater. 2006, 38, 253–267.
- Cadoni, E.; Solomos, G.; Albertini, C. Concrete behaviour in direct tension tests at high strain rates. Mag. Concr. Res. 2013, 65, 660–672.
- Thomas, R.J.; Sorensen, A.D. Review of strain rate effects for UHPC in tension. Constr. Build. Mater. 2017, 153, 846–856.
Point 6: Page 7 Line 227-241
This whole part should be in the introduction, does not make sense in the results section, where you should be discussing your results.
Response 6: Thanks for your comment. We accepted the reviewer's comments and moved these sentences to the introduction. Also, the restructuring brought about a change in the order of references is modified in the manuscript.

Reviewer 3 Report
In this paper, the mechanical behaviors of concrete in dynamic splitting test are investigated. The stress wave signals collected by dynamic strain gauge are analyzed, and the variation of the concrete dynamic tensile strength against the stress rate is observed. The strain field on the surface of concrete specimens is studied. However, there are several studies on the failure characteristic of concrete using dynamic splitting tests. The novelty of the study has not been presented and needs to be clarified. Other comments are as follows.
-There are some grammatical errors. For instance Line 88: by combing the high-speed camera and the DIC technology. Please correct.
-Conclusions need to present the key results and also highlight any recommendations drawn by this study. The conclusions are quite similar to other researchers. Please highlight and clarify the recommendations in this paper.
-What do the authors mean on deformation characteristics of concrete, it needs to be clarified further in the manuscript.
Author Response
Response to Reviewer 3 Comments
Dear Editors and Reviewers:
Thanks for your letter and for the reviewers' comments concerning our manuscript entitled "Investigation on the deformation and failure characteristics of concrete in dynamic splitting tests" (ID: materials-1594639). The comments are all valuable and helpful for revising and improving our paper, as well as the important guiding significance to our research. We have studied the comments carefully and have made corrections which we hope are met with approval. Revised portions are marked in red in the paper (Track Changes Version). The main corrections in the paper and the response to the reviewers' comments are as following:
In this paper, the mechanical behaviors of concrete in dynamic splitting test are investigated. The stress wave signals collected by dynamic strain gauge are analyzed, and the variation of the concrete dynamic tensile strength against the stress rate is observed. The strain field on the surface of concrete specimens is studied. However, there are several studies on the failure characteristic of concrete using dynamic splitting tests. The novelty of the study has not been presented and needs to be clarified. Other comments are as follows.
Point 1: There are some grammatical errors. For instance, Line 88: by combing the high-speed camera and the DIC technology. Please correct.
Response 1: Thanks for you point out the grammatical errors. The sentence at line 88 was modified as: "The strain field of concrete specimens at the initiation of cracking obtained by high-speed image analysis with the DIC technique."
Point 2: Conclusions need to present the key results and also highlight any recommendations drawn by this study. The conclusions are quite similar to other researchers. Please highlight and clarify the recommendations in this paper.
Response 2: Thanks for your comment. We added some sentences to the first paragraph of the conclusion to highlight and clarify the recommendations of this paper. "Based on the experimental results the incubation times of concrete specimens with 100 mm diameter and 50 mm high were determined, the deformation characteristics of the cracking initiation stage were revealed by the DIC technique, and the dynamic splitting failure process was described by the Gaussian function. The detailed conclusions are as follows:"
Point 3: What do the authors mean on deformation characteristics of concrete, it needs to be clarified further in the manuscript.
Response 3: Thanks for your comment. The deformation characteristics of concrete in this paper refer to the full-field strain distribution characteristics of the specimen at the initiation of cracking. To highlight this, we modified the title of Section 3.3 as "Deformation characteristics of concrete measured by DIC".
